# Pacing in World-Class Age Group Swimmers in 100 and 200 m Freestyle, Backstroke, Breaststroke, and Butterfly

**DOI:** 10.3390/ijerph17113875

**Published:** 2020-05-30

**Authors:** Cathia Moser, Caio Victor Sousa, Rafael Reis Olher, Pantelis Theodoros Nikolaidis, Beat Knechtle

**Affiliations:** 1Balgrist University Hospital, 8008 Zurich, Switzerland; cathiamoser@gmx.ch; 2Bouve College of Health Sciences, Northeastern University, Boston, MA 02115, USA; cvsousa89@gmail.com; 3Programa de Pós-Graduação Stricto Sensu em Educação Física, Universidade Católica de Brasília, Brasília-DF 72445-020, Brazil; rflolher@gmail.com; 4Exercise Physiology Laboratory, 18450 Nikaia, Greece; pademil@hotmail.com; 5School of Health and Caring Sciences, University of West Attica, 12243 Athens, Greece; 6Institute of Primary Care, University Hospital Zurich, 8091 Zurich, Switzerland; 7Medbase St. Gallen Am Vadianplatz, 9001 St. Gallen, Switzerland

**Keywords:** world class, age-group, pacing, performance, sex difference

## Abstract

Pacing in swimming has been investigated in pool swimming for elite-standard and age group freestyle swimmers, but little is known about pacing in age group swimmers competing at world class level in backstroke, breaststroke, and butterfly. The aim of this study was to investigate pacing for age group swimmers competing at world class level in 100 and 200 m in the four single disciplines (freestyle, backstroke, breaststroke and butterfly). Data on 18,187 unique finishers competing in four FINA Master World Championships between 2014 and 2019 were analyzed. The sample included 3334 women and 14,853 men. Swimming speed decreased with increasing age (*p* < 0.05). Freestyle was the fastest and breaststroke the slowest (*p* < 0.05) stroke. Women and men were faster in 100 m (*p* < 0.05) than in 200 m. Backstroke was the stroke with the lowest and butterfly with the highest coefficient of variation in swimming speed. One hundred meters had a higher coefficient of variation in swimming speed than breaststroke (*p* < 0.05). For 100 m, swimming speed decreased for all strokes and all age groups during the second lap. For 200 m, swimming speed was the fastest for all strokes and all age groups during the first lap. In summary, the FINA World Masters Championships presented the unique characteristic that, when all competitors were considered, (i) swimming speed decreased with increasing age, (ii) women and men were faster in 100 m than in 200 m, (iii) freestyle was the fastest stroke and (iv) the largest increase in swimming time for 100 m all strokes and all age groups occurred during the second (out of two) lap and for 200 m, swimming speed was the fastest for all strokes and age groups during the first lap. These findings should help coaches to develop age- and event-tailored pacing strategies.

## 1. Introduction

Master athletes can be considered as a model of successful ageing because they provide a unique opportunity to study human physical performance potential; therefore, there is an increasing scientific interest in them [1,2]. One of the most popular sports disciplines is swimming with many elderly athletes practicing it at a recreational level and a large number of master athletes participating in official sport events such as the FINA (Fédération Internationale de Natation) World Masters Championship [3]. 

Participation and performance trends for master swimmers have been investigated for all four single disciplines of freestyle [4], butterfly [5], breaststroke [6] and backstroke [7]. Participation in master freestyle swimmers increased from 1986 to 2014 in women and men in older age groups. Moreover, both women and men improved their performance in all distances across time, and women were not slower compared to men in age groups 80–84 to 85–89 years. 

Pacing—in the most simplified terms—can be defined as the distribution of exercise intensity during any kind of a race [8,9]. Abbiss and Laursen [10] described six pacing strategies in athletic performance such as negative pacing (i.e., increase in speed over time), positive pacing (i.e., continuous slowing over time), all-out pacing (i.e., maximal speed possible), even pacing (i.e., same speed over time), parabolic-shaped pacing (i.e., positive and negative pacing in different segments of the race) and variable pacing (i.e., pacing with multiple fluctuations). In masters swimming, pacing has been investigated for master freestyle swimmers for all distances [11] but not for master swimmers of other disciplines and distances. For age group freestyle swimmers in the events of 100, 200, 400 and 800 m, the largest increase in swimming time occurred during the second lap, and the largest decreases in swimming time occurred during the last lap, except in the event of 100m [11]. However, no data exist about pacing in age group swimmers in backstroke, breaststroke and butterfly. 

Age has been reported as an important variable in pacing of endurance athletes such as marathon runners [12]. For age group freestyle swimmers, it has been shown that the effect of age group was greater than the effect of participants’ sex, and women were not slower compared to men in age groups 80–84 to 85–89 years in the FINA World Masters Championships [11]. However, no study investigated the aspect of age in pacing in age group swimmers for other disciplines such as backstroke, breaststroke and butterfly. The knowledge of pacing in age group swimmers for backstroke, breaststroke and butterfly might have practical implications for both sports scientists and coaches working with age group swimmers competing in other strokes than butterfly at world class level. Information about changes in swimming time by laps in age group swimmers would help to design specific age-tailored training programs. 

The present study investigated changes in swimming time by laps in age group swimmers competing in the FINA World Masters Championships 2014, 2015, 2017 and 2019 in 100 and 200 m freestyle, backstroke, breaststroke and butterfly. The hypothesis was that older swimmers would slow down with increasing age and with increasing distance without an impact of sex and discipline.

## 2. Materials and Methods 

### 2.1. Ethics Approval

This study was approved by the Institutional Review Board of St. Gallen, Switzerland, with a waiver of the requirement for informed consent of the participants as the study involved the analysis of publicly available data (EKSG 01-06-2010). 

### 2.2. Data Sampling and Data Analysis

All data were obtained from the official and publicly accessible website of the FINA [13] on 1 February 2020. Any swimmer older than 25 years fulfilling the qualification time and affiliated to an official swimming club can start in a FINA World Masters Championship [14]. Trial times for 100 m and 200 m distances, in 50 m increments, were recorded in the XV FINA World Masters Championships held in Montreal, Canada, in 2014; in the XVI FINA World Masters Championships held in Kazan, Russia, in 2015; in the XVII FINA World Masters Championships held in Budapest, Hungary, in 2017 and in the XVIII FINA World Masters Championships held in Gwangju, South Korea, in 2019. A total of 4857 women and 6239 men swimmers who competed in 100 m and 3334 women and 3753 men who competed in 200 m freestyle, backstroke, breaststroke and butterfly were considered. We included all women and all men from every 5-year age groups from 25–29 years to 95–99 years to avoid a selection bias by analyzing only a limited sample of top athletes such as the top 10 or top 100 of each age group.

### 2.3. Statistical Analyses

All statistical procedures were carried out using the Statistical Package for the Social Sciences (SPSS version 26. IBM, NY, USA) and GraphPad Prism (version 8.4.2. GraphPad Software LLC, CA, USA). Based on lap time, the coefficient of variance (CV) in lap times was calculated for each participant as CV = (standard deviation/mean) × 100. Lap times and CV were the dependent variables, whereas sex, age group, stroke and distance were defined as independent variables. The Shapiro–Wilk and Levene’s tests were applied for normality and homogeneity, respectively. General linear models were applied with Distance × Sex × Age group × Stroke as factors. Sex was always included as fixed factor while all other variables were analyzed as random factors. When interactions were found (*p* < 0.05), pairwise comparisons were applied to identify the differences more accurately. The hypothesis of sphericity was verified by Mauchly test, and when violated, the degrees of freedom were corrected by the Greenhouse–Geisser estimates. The level of significance utilized was *p* ≤ 0.05.

A positive pacing strategy is observed when the speed gradually declines throughout the duration of the event whereas a negative pacing strategy is observed when there is an increase in speed over the duration of the event. In an all-out pacing strategy, speed tends to gradually decrease, possibly resulting in suboptimal performance times after an athlete has reached peak velocity [10].

## 3. Results

Participants from 100 m (*n* = 11,100) and 200 m (*n* = 7087) were included, for a total sample size of 18,187 age group athletes with 3334 women and 14,853 men. The four strokes comprised data for freestyle (*n* = 6729), butterfly (*n* = 2606), backstroke (*n* = 3547) and breaststroke (*n* = 5332) (Table 1). Finally, the age groups were separated by sex, distance and stroke; the age group with the lowest number of participants was women in the age group ≥70 years competing in the 200 m butterfly, with only 36 athletes. The age group with the highest number of participants was 100 m freestyle of men 50–54 years old (M55) with 358 athletes.

The multifactorial model for average swimming speed showed a trend towards significance (*p* = 0.059) for sex, and significant effects (*p* < 0.05) for age group, stroke, distance and the interactions sex × age group, sex × distance, age group × stroke, age group × distance, stroke × distance, sex × age group × stroke and sex × stroke × distance (Table 2). Pairwise comparisons showed that swimming speed decreased with increasing age, where the 25–29 and 30–34 were the fastest age groups for both women and men among all strokes, except 200 m butterfly, where 30–34 was the fastest age group in both men and women. Freestyle was the fastest stroke and breaststroke the slowest. Women and men were faster in 100 m than in 200 m (Figure 1).

The multifactorial model for the individual CV showed a significant effect (*p* < 0.05) for stroke, a trend (*p* = 0.054) for the interaction sex × age group and significant effect for the interactions sex × stroke, age group × stroke, stroke × distance and sex × stroke × distance (Table 2). Pairwise comparisons showed that backstroke was the stroke with the lowest CV and butterfly the stroke with the highest CV. One hundred meters showed the highest CV in breaststroke (Figure 2).

The multifactorial model for pacing in 100 m races (Figure 3a–d) showed a significant effect for swim stroke (F = 1385.4; *p* < 0.001), sex (F = 2215.7; *p* < 0.001), pace (F = 67,268.7; *p* < 0.001) and for the interactions sex × stroke (F = 14.9; *p* < 0.001) and pace × stroke (F = 1258.5; *p* < 0.001). The multifactorial model for pacing in 200 m races (Figure 3e–h) showed a significant effect for swim stroke (F = 527.7; *p* < 0.001), sex (F = 851.1; *p* < 0.001), pace (F = 18,474.1; *p* < 0.001) and for the interactions sex × stroke (F = 9.3; *p* < 0.001) and pace × stroke (F = 382.4; *p* < 0.001).

## 4. Discussion

This study investigated for the first time pacing in world-class age group swimmers in 100 and 200 m freestyle, backstroke, breaststroke and butterfly with the hypothesis that older swimmers would slow down with increasing age and with increasing distance without an impact of sex and discipline. Based on the current results for 400 and 800 m freestyle, master swimmers adopted a parabolic pacing [10], and the first and the last lap were the fastest. The same pacing strategy was also observed in elite-standard swimmers competing in 400 m freestyle swimming where a fast-start-even-and-parabolic pacing strategy was used [15].

Based on our results, the main findings were: (i) average swimming speed decreased with increasing age, where M30 and F30 were the fastest for both women and men among all styles, except 200 m butterfly, where M35 and F35 were the fastest age groups in both men and women; (ii) freestyle was the fastest stroke and breaststroke the slowest; (iii) women and men are faster in 100 m races than in 200 m, (iv) the first lap was the fastest one for 100 m and 200 m, and (v) for 200 m, the largest increase during swimming time for all age groups and for all strokes occurred during the second lap (out of four).

A first finding was that swimming speed decreased with increasing age, but the youngest age group (25–29 years) was not the fastest age group. It might be possible that the world class elite swimmers do not swim anymore after their career has been finished, and in the masters swimmers, the second age group swimmers begin by competing. This study found that the age group 30–34 years was the fastest for both women and men in all strokes except 200 m butterfly, where the age group 35–39 years was the fastest in both men and women. Similar findings have been reported for other sports disciplines such as running for example in triathlon races [16].

Regarding the marathon races in Oslo from 2008 to 2018, the fastest men and women runners were also in the age group 35–39 years, whereas the oldest runners were the slowest [17]. In pacing of Ironman triathletes, it has previously been shown that the younger age groups were relatively faster in swimming than their older counterparts [18]. However, the youngest age group is not the fastest in achieving peak performances for top athletes. The age of peak performance for athletes specializing in specific events can be obtained from the equations of the linear trends [19]. Apart from age, sex is another important variable regarding pacing in athletes. To date, we have knowledge about the sex differences in pacing during half-marathon and marathon running [20]. Men marathon runners showed greater speed fluctuations than women, whereas in half-marathon, both men and women had rather similar pacing profiles. In addition, success does not only result from age but highlights also the importance for example of the oxygen uptake, maximal heart rate, stroke volume, arteriovenous oxygen difference, active muscle mass, type II muscle fibre size and blood volume. In masters endurance athletes, type I fiber composition is increased, whereas type II fiber size is reduced. This may have a positive impact on muscle capillarization, oxidative metabolism and therefore endurance performance [21].

A second finding was that freestyle was the fastest style and breaststroke the slowest. Another study analyzing elite athletes in individual medley has shown that men accomplished a lower race time in breaststroke than did the women and a faster race time in the freestyle in both the 200- and 400-m distances, with the fastest stroke for both sexes being the butterfly. Considering only the medalists in elite athletes, in men (200 and 400 m), the backstroke was the stroke that most determined their final performance, whereas in women, it was the backstroke (200 m) or freestyle (400 m) [22].

A third finding was that women and men were faster in the 100 m races than in the 200 m. This trend has been previously observed in analyses of successful pacing profiles of Olympic and IAAF World Championship Middle-Distance runners competing in 800 m races. The fastest speeds were achieved over the first 200 m followed by a considerable decrease in pace to 300 m, with pace maintained to 500 m [23]. This finding might be interpreted considering the different human energy transfer systems engaged in performance as the distance of the race increased [24]. Compared to the shortest distance relied mostly on anaerobic system, the longer the race, the more the resynthesis of ATP relied on aerobic system; althought the aerobic system allowed swimimming for longer distance, its power was smaller than anaerobic system.

A fourth finding was that the largest increase in swimming time for 100 m for all age groups and all strokes occurred during the second lap (out of two) (positive pacing). This trend has been previously observed in analyses of top 16 finishers (semi-finalists, finalists) in nine international competitions over a 7-year period. Finalists exhibited very large correlations with final time in the second 50-m lap of 100 m events [25]. An increase of swimming time during the second half of 100 m race indicated a positive pacing. It might be assumed that the accumulated fatigue (increase of blood lactate, metabolic acidosis) caused by high-intensity (predominantly anaerobic) [26] exercise such as the performance in the first half of 100 m race did not allow the maintenance of swimming speed throughout the race leading into an increase of swimming time.

A last important finding was that for 200 m, the largest increase during swimming time for all age groups and for all strokes occurred during the second lap (out of four) (positive pacing). Another study analyzing the relationships between pacing parameters and performance of elite 1500 m swimmers found that most of the reduction in time of the first lap was likely a consequence of the dive and underwater glide following the starting signal, but the second lap was also faster on average than the time predicted by the linear and quadratic parameters, suggesting a contribution of a relatively faster swimming speed in the first lap [27]. An interpretation of this finding might be that the ability of human body for muscle as a function of time would follow a parabolic (half “U-shape”) rather than a liner relationship. Accordingly, swimming time would increase (i.e., get slower) more in the second lap—compared to the first—than in the last two laps. Considering their physiological relevance, it might be assumed that the second, third and fourth laps (all with high contribution of the aerobic energy transfer system) would have larger affinity among them than with the first lap (relying mostly on the anaerobic system) [28].

With regards to sex difference in pacing, another study analyzed pacing strategies in 200 and 400 m individual medley in international swimming competitions and found that, in general, men adopted faster swimming speeds over the first half of the races (positive pacing) in the 200 and 400 m individual medley events, whereas in the last lap, the women swam faster than men did (negative pacing) [22]. In the analysis of lap times in international swimming competitions, the laps with the strongest relationship to final time were the final lap for sprint events and the middle two laps for 200- and 400-m events [25]. Analysis of the pacing parameters of the best swims in pacing profiles and competitive performance of elite female 400-m freestyle swimmers showed that the lap-to-lap variability was the lowest for the best swims of the record holders, who also displayed slower first and faster last laps [29]. A low lap-to-lap variability might be considered relevant to a more even pacing, characteristic of high-performance level athletes showing an ability to distribute energy optimally throughout a race. On the other hand, an incorrect distribution of energy might lead athletes of lower performance level either to start fast and show large decrease of speed (positive pacing) or to start slow and increase speed in the last part of a race (final spurt).

In addition, race success does not only result from age but highlights also the importance of physiological variables such as oxygen uptake, maximal heart rate, stroke volume, arteriovenous oxygen difference, active muscle mass, type II muscle fiber size and blood volume [21,30]. Maximum oxygen consumption (VO_2_max) is not only an indicator of endurance performance but also a strong predictor of cardiovascular disease and mortality. This physiological parameter is known to decrease with aging [31]. The pacing in these master swimmers might be explained by physiological aspects. The gradual loss of muscle mass and strength is one of the more consistent hallmarks of normal aging. It has been well-established that sarcopenia is characterized by a decline in skeletal muscle mass and strength and the loss of functional capacity with aging [32]. Among the three main physiological determinants of endurance exercise performance (i.e., VO_2_max, lactate threshold and exercise economy), a progressive reduction in VO_2_max appears to be the primary mechanism associated with declines in endurance performance with age. A reduction in lactate threshold, i.e., the exercise intensity at which blood lactate concentration increases significantly above baseline, also contributes to the reduction in endurance performance with ageing, although this may be secondary to decreases in VO_2_max [1]. Analysis of a competitive 100 m freestyle in elite male swimmers showed that the relative alactic contribution decreased, the aerobic contribution increased up to 100 m, and the lactic anaerobic contribution remained stable. Higher VO_2_, blood lactate concentration, alactic and lactic energy expenditures were exhibited by older swimmers [33]. This can be an explanation why older athletes decrease in swimming time during the second lap.

Some limitations must be acknowledged for this study. In the FINA Masters World Championships, freestyle swimmers can be considered as the best age group swimmers in the world. In lower level swimmers there is a chance for a different pacing; therefore, the pacing patterns should be generalized for swimmers of a similar level. It is also important to note that the competition took place in a 50-m-long outdoor pool with different physiological aspects (e.g., turns or push parts off the wall) compared to a smaller pool with for example 25 m in length. Another limitation of this study might be the environmental conditions such as temperature for air and water because the competition did not take place indoors. Finally, a statistically significant outcome was clearly attributable to the high number of participants.

Similar analyses (performance and pacing across calendar years) were done in several endurance sports and reported interesting results. However, we believe for a “calendar/ event analysis” more years should be on the sample. Thus, for this particular study, we decided to consider and analyze the distance and style effect over pacing and performance. For future studies, it would be interesting to investigate differences between specific events.

## 5. Conclusions

In summary, the average swimming speed decreased with increasing age, where athletes in the age group 30–35 years were the fastest for both women and men among all styles, except for 200 m butterfly, where athletes in the age group 35–39 were the fastest in both men and women. Freestyle was the fastest style and breaststroke the slowest. Women and men are faster in 100 m races than in 200 m ones. In 100 m, the largest increase in swimming time occurred during the second lap (out of two) (positive pacing) for all age groups and all strokes. In 200 m, the largest increase during swimming time for all age groups and for all strokes occurred during the second lap (out of four) (positive pacing).

## Figures and Tables

**Figure 1 ijerph-17-03875-f001:**
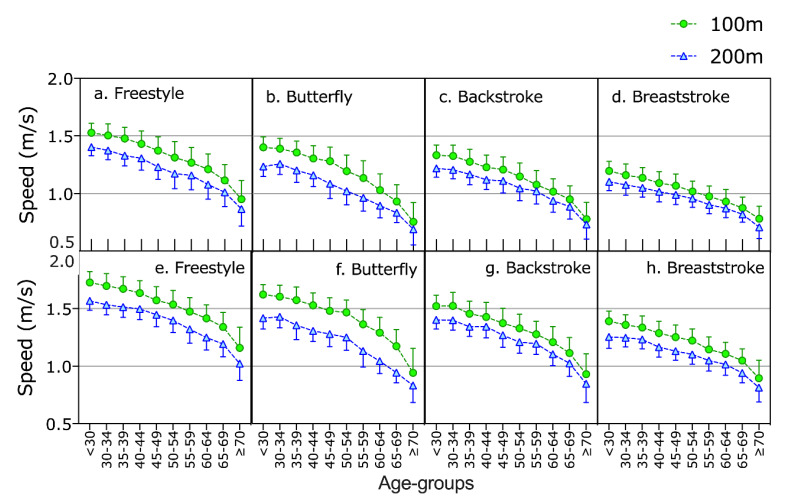
Swimming speed of women (**a**–**d**) and men (**e**–**h**) in 100 m and 200 m of the four strokes across age groups.

**Figure 2 ijerph-17-03875-f002:**
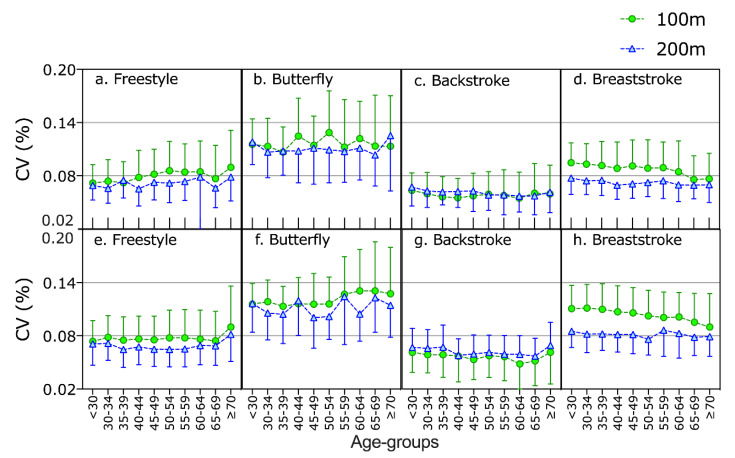
Individual coefficient of variance of swimming speed of women (**a**–**d**) and men (**e**–**h**) in 100 m and 200 m swim races of the four strokes across age groups.

**Figure 3 ijerph-17-03875-f003:**
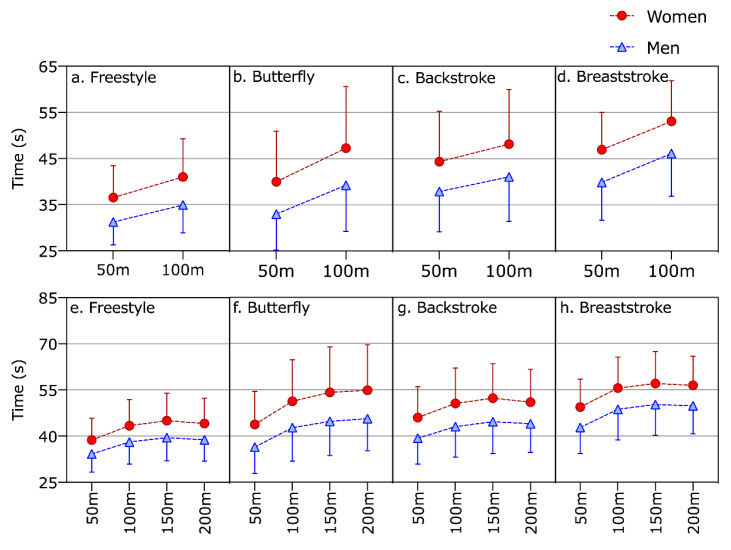
Race time for each 50 m split in 100 m (**a**–**d**) and 200 m (**e**–**h**) of men and women in the four strokes.

**Table 1 ijerph-17-03875-t001:** Number of athletes adopting a positive or negative pacing in 100m and 200m swim races in the four strokes.

Distance	Stroke	Positive Pacing	Negative Pacing
100 m	Freestyle	4221	13
	Butterfly	1635	1
	Backstroke	1888	26
	Breaststroke	3243	5
200 m	Freestyle	2443	1
	Butterfly	940	0
	Backstroke	1569	2
	Breaststroke	2074	0

**Table 2 ijerph-17-03875-t002:** Multifactorial model with dependent variables being average swimming speed and coefficient of variance (CV).

Factor	Average Speed		*CV*	
	F	*p*-Value	F	*p*-Value
Sex	114.2	0.059	9.3	0.202
Age group	446.2	<0.001	1.3	0.359
Stroke	61.5	0.003	17.1	0.022
Distance	26.4	0.014	2.5	0.209
Sex × Age group	5.4	0.010	3.1	0.054
Sex × Stroke	3.5	0.164	11.2	0.039
Sex × Distance	14.1	0.026	0.8	0.429
Age group × Stroke	13.9	<0.001	4.0	<0.001
Age group × Distance	4.0	0.015	1.8	0.175
Stroke × Distance	14.3	0.017	15.4	0.013
Sex × Age group × Stroke	2.3	0.015	1.8	0.070
Sex × Age group × Distance	1.3	0.258	0.9	0.513
Sex × Stroke × Distance	5.9	0.003	4.6	0.009
Distance × Age group × Stroke	1.7	0.096	1.6	0.101
Distance × Sex × Age group × Stroke	1.0	0.458	1.3	0.136

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
