# Peer review of "Pacing in World-Class Age Group Swimmers in 100 and 200 m Freestyle, Backstroke, Breaststroke, and Butterfly"

_ijerph, 2020, doi:10.3390/ijerph17113875_

Round 1

Reviewer 1 Report

The article seems good, easy and applicable in general terms. Also is a good point to analyze all swimming styles.

 However I have some questions / suggestions for authors in order to improve the paper and / or increase its applicability.

  1. Main results and conclusions were predictable just been in a swimming pool working with swimmers and a stopwatch. So, What is new?

  1. Authors discuss their results according to other sports like triathlon, running… but it is not the same events of 1-2’ of duration than other events such as half marathon / marathon or ironman (hours) .So it will be interesting to compare data with swimming events. What about elite swimmers vs masters swimmers? Do they have same pacing?

  1. One of the practical application derived from this paper is for coaches… What authors suggest for training in each swimming event for masters? Winners (or medalists or finalist) have the same pace that other ones?

  1. In introduction section, authors described (Abbiss and Laursen) six pacing strategies, however in results sections they only report 2 strategies: positive and negative. There were not any all out? Which were criteria for defining how a pacing strategy was considered positive, negative, all out… in this paper. It should be added y methods sections.

  1. Another interesting point would be to know if there were differences between different World Championship. I.e: Montreal pacing was different from Gwanju? Do Pacing strategies maintain stables during years or they changes after years….

  1. Why 25-29 age group are not fasters ones? It should explained better and in a swimming context. Is it possible that some of these swimmers are competing in elite or they have just finished their careers as swimmers and need “some time” to return to swimming pools?. Please, explore this.

  1. Regarding to paragraph in discussion section (lines 228-238), it should be considered an important physiological aspect such as anaerobic Lactate production and accumulation is very important for performance in 100m-200m  swimming events. Please add more information about this in discussion. I.e: lactate values in 100-200m in swimmers and how anaerobic metabolism decrease with age.

Author Response

Reviewer 1

The article seems good, easy and applicable in general terms. Also is a good point to analyze all swimming styles. However, I have some questions / suggestions for authors in order to improve the paper and / or increase its applicability. Main results and conclusions were predictable just been in a swimming pool working with swimmers and a stopwatch. So, what is new?

Answer: We agree with the expert reviewer and thank you for the input. Our intention was to present a paper showing that there is no data about pacing in world-class age group swimmers in freestyle, backstroke, breaststroke and butterfly. With our paper we were able to fill this gap in science.

Authors discuss their results according to other sports like triathlon, running… but it is not the same events of 1-2’ of duration than other events such as half marathon / marathon or ironman (hours). So it will be interesting to compare data with swimming events. What about elite swimmers vs masters swimmers? Do they have same pacing?

Answer: We agree with the expert reviewer. We edited it in the text. Based on the current results for 400 m and 800 m freestyle, master swimmers adopted parabolic pacing (Abbissand Laursen, 2008) and the first and the last lap were the fastest. The same pacing strategy was also observed in elite-standard swimmers competing in 400 m freestyle swimming where a fast-start-even-and-parabolic pacing strategy was used (Mauger et al.2012).

One of the practical application derived from this paper is for coaches… What authors suggest for training in each swimming event for masters? Winners (or medalists or finalist) have the same pace that other ones?

Answer: We agree with the expert reviewer and thank you for the indication.

To illustrate this point, we have knowledge that exercise performance can be significantly

influenced by the distribution of work during an exercise task. However, the precise pacing

strategies that ensure the best possible performance outcome under the variety of existing

athletic competitions are not clear. Evidence suggests thatduring events, well trained athletes

tend to adopt a positive pacing strategy but after the peak speed is reached, the athlete

progressively slows. The underlying mechanisms influencing the regulation of pace during

exercise are still unclear. During swimming the dive start allows athletes to reach

maximal velocity within a relatively short time. Similarly, the ‘flying start’ that occurs during

the final three legs of relay athletic races reduces the influence of acceleration on overall

pacing strategy. Athletes and therefore not only winners adopt a positive pacing strategy

during such events.

In introduction section, authors described (Abbiss and Laursen) six pacing strategies, however in results sections they only report 2 strategies: positive and negative. There were not any all out? Which were criteria for defining how a pacing strategy was considered positive, negative, all out… in this paper. It should be added y methods sections.

Answer: We agree with the expert reviewer. A positive pacing strategy is observed when the

speed gradually declines throughout the duration of the event whereas a negative pacing

strategy is observed when there is an increase in speed over the duration of the event. In an all-

out pacing strategy, speed tends to gradually decrease, possibly resulting in suboptimal

performance times after an athlete has reached peak velocity. In addition, there are U, J or

reverse J-shaped pacing strategies, but those strategies are the result of athletes adopting

both, a positive and negative pacing strategy during an event. Based on our current results the swimming time increased in 100 m and 200 m during the second lap which can be called as a positive pacing.

Another interesting point would be to know if there were differences between different World Championship. I.e: Montreal pacing was different from Gwanju? Do Pacing strategies maintain stables during years or they change after years….

Answer: We agree with the expert reviewer. We actually have done similar analysis (performance and pacing across calendar years) in several endurance sports and reported interesting results. However, we believe that a for a “calendar/event analysis” more years should be on the sample. Thus, for this particular study we decided to consider and analyze the distance and style effect over pacing and performance.

Why 25-29 age group are not fasters ones? It should explained better and in a swimming context. Is it possible that some of these swimmers are competing in elite or they have just finished their careers as swimmers and need “some time” to return to swimming pools?. Please, explore this.

Answer: We agree with the expert reviewer. It might be possible that the world class elite swimmers don’t swim anymore after their career has been finished and in the master swimmers the second age group swimmers begin with competing.

Regarding to paragraph in discussion section (lines 228-238), it should be considered an important physiological aspect such as anaerobic Lactate production and accumulation is very important for performance in 100m-200m swimming events. Please add more information about this in discussion. I.e: lactate values in 100-200m in swimmers and how anaerobic metabolism decrease with age.

Answer: We agree with the expert reviewer. Anaerobic qualities become more important with

age. Analysis of a competitive 100 m freestyle in elite male swimmers showed that the relative

alactic contribution decreased, the aerobic contributionincreased up to 100 m and the lactic

anaerobic contribution remained stable. Higher VO2 blood lactate concentration, alactic and

lactic energy expenditures were exhibited by older swimmers.

Reviewer 2 Report

The authors are commended for their analysis and manuscript. Though the overall message is clear and data-driven, a few revisions remain. Some general comments are: be consistent with terminology and notation (e.g. used “world-class” as opposed to “world class”). Lastly, what are the future steps in this area of research?

Please see below for specific comments. All the Best.

Specific comments

Page 1, lines 38 – 39: The opening message is there, but revise for readability. At the moment it reads choppy.

Page 2, line 44: Place “of” between “disciplines” and “freestyle”.

Page 2, lines 45 – 47: Break-up into to separate sentences. For the first, include data to illustrate the increase in participation from 1986 to 2014. In the second, include data to illustrate improvements and comparisons. Consider something as simple as percent changes and differences.

Page 2, line 71: Place a space between “2015,” and “2017”.

 Page 2, line 71: Omit “in” after “200 m”.

Page 2, lines 72 – 74: Separate the hypothesis statement into a separate/independent sentence.

Page 2, lines 75 – 79: Not sure why ethics was required if the authors are simply analyzing data available to the public.

Page 2, lines 82 – 83: Please note the date or dates (date(s), month(s) and year(s)) that the data was retrieved from the FINA website.

Page 2, line 86: Remove the parentheses about “Canada”, and write as “Montreal, Canada”. Repeat for the remaining locations.

Page 2, lines 86 – 90: Simply place “Trial times for 100 and 200 m distances, in 50 m increments, were recorded” before “for the XV FINA World Masters Championships…”. This will increase readability and clarity.

Page 3, line 94: Place a space in line 95.

Page 3, line 98: Place “(CV)” after “variance”. Also, note how CV was calculated as there are a couple accepted methods in the literature.

Page 3, line 102: Why was “sex × swim” analyzed twice? Also, place “while” after “factor,” add “variables” after “other”, and replace “included” with “analyzed”.

Page 3, line 112: what is meant by “and above with 36 athletes”? Revise for clarity.

Page 3, line 118: Place “towards” between “trend” and “significance” for readability.

Page 3, lines 122 and 123: What does “M30”, “F30”, “M35: and “F35” stand for? Please note before abbreviating.

Table 2: Place entire table on Page 4 for readability.

Figure 1: Please define the “M” groups for clarity. This defining will probably be more appropriate in the text, as opposed to under the figure description.

Page 4, lines 133 and 134: Be consistent with how interactions are noted. Previously it was without spaces about the “×”; here there are spaces. The latter is encouraged.

Table 3: same as Table 2.

Page 5, line 141: Place a space between the Figure 2 description and following text.

Page 5, line 142: Place a space between :100” and “m” for consistency.

Page 6, line 168: Specifically define “running”. Also, please provide references to support the statement.

Page 7, lines 178 – 180: What does the literature say about Type I fibre types; being that they are more aerobic in nature? Also see Page 8, line 223.

Page 7, lines 183 – 184: What is meant by “employed a smaller percentage of their event times”? Please clarify. Same for following statement.

Page 7, lines 190 – 194: Expound more on the ‘why’ behind this. Yes, it may be considered intuitive, but there are factors (chiefly physiological and psychological factors) that are playing a large role in this observation.

Page 7, lines 196 – 200: Again, ‘why’? What are the authors speculations about this observation?

Page 7, lines 202 – 208: Same as above.

Page 7, line 210: Replace “further” with “another” for readability.

Page 7, lines 210 – 219: Same as above. What are the authors thoughts on why we are seeing what we are seeing?

Page 8, line 235: Eliminate space and hyphen in “baseline”.

Author Response

Reviewer 2

The authors are commended for their analysis and manuscript. Though the overall message is clear and data-driven, a few revisions remain. Some general comments are: be consistent with terminology and notation (e.g. used “world-class” as opposed to “world class”). Lastly, what are the future steps in this area of research?

Please see below for specific comments. All the Best.

Specific comments

Page 1, lines 38 – 39: The opening message is there, but revise for readability. At the moment it reads choppy.

Answer: We agree with the expert reviewer and rephrase the text in the manuscript. Master athletes can be considered as a model of successful ageing because they provide a unique opportunity to study human physical performance potential, therefore it exists an increasing scientific interest. We hope now there is not any misunderstanding.

Page 2, line 44: Place “of” between “disciplines” and “freestyle”.

Answer: We agree with the expert reviewer and changed as suggested.  

Page 2, lines 45 – 47: Break-up into to separate sentences. For the first, include data to illustrate the increase in participation from 1986 to 2014. In the second, include data to illustrate improvements and comparisons. Consider something as simple as percent changes and differences.

Answer: We agree with the expert reviewer and the paragraph was adjusted as follows: "Participation and performance trends for master swimmers have been investigated for all four single disciplines of freestyle [4], butterfly [5], breaststroke [6] and backstroke [7]. Participation in master freestyle swimmers increased from 1986 to 2014 in women and men in older age groups. Moreover, both women and men improved their performance in all distances across time, and women were not slower compared to men in age groups 80–84 to 85–89 years." 

Page 2, line 71: Place a space between “2015,” and “2017”.

Answer: We agree with the expert reviewer and changed as suggested.

Page 2, line 71: Omit “in” after “200 m”.

Answer: We agree with the expert reviewer. We revised it.

Page 2, lines 72 – 74: Separate the hypothesis statement into a separate/independent sentence.

Answer: We agree with the expert reviewer and separated the hypothesis statement into an independent sentence.

Page 2, lines 75 – 79: Not sure why ethics was required if the authors are simply analyzing data available to the public.

Answer: We agree with the expert reviewer but we want to mention that some journals request an approval of an ethics committee, even if there was simply analyzing data available to the public. Therefore it is necessary for us that we inform in the manuscript about ethics approval.

Page 2, lines 82 – 83: Please note the date or dates (date(s), month(s) and year(s)) that the data was retrieved from the FINA website.

Answer: We agree with the expert reviewer and changed as suggested.

Page 2, line 86: Remove the parentheses about “Canada”, and write as “Montreal, Canada”. Repeat for the remaining locations.

Answer: We agree with the expert reviewer. We edited it in the text.

Page 2, lines 86 – 90: Simply place “Trial times for 100 and 200 m distances, in 50 m increments, were recorded” before “for the XV FINA World Masters Championships…”. This will increase readability and clarity.

Answer: We agree with the expert reviewer and changed as suggested.

Page 3, line 94: Place a space in line 95.

Answer: We agree with the expert reviewer. We edited it in the text.

Page 3, line 98: Place “(CV)” after “variance”. Also, note how CV was calculated as there are a couple accepted methods in the literature.

Answer: We agree with the expert reviewer. The “CV” was moved and the equation used to calculate CV was added, as requested as follows: CV=(standard deviation/mean)*100.We hope now there is not any misunderstanding.

Page 3, line 102: Why was “sex × swim” analyzed twice? Also, place “while” after “factor,” add “variables” after “other”, and replace “included” with “analyzed”.

Answer: It was an error with factors included. It was rephrased. Thank you for the comment. We hope now there is not any misunderstanding.

Page 3, line 112: what is meant by “and above with 36 athletes”? Revise for clarity.

Answer:We agree with the expert reviewer and it was rephrased as follows: "Finally, the

age groups were separated by sex, distance and stroke; the age group with the lowest number

of participants was women in the age group ≥70 racing the 200 m butterfly, with only 36

athletes." 

Page 3, line 118: Place “towards” between “trend” and “significance” for readability.

Answer: We agree with the expert reviewer and changed as suggested.

Page 3, lines 122 and 123: What does “M30”, “F30”, “M35: and “F35” stand for? Please note before abbreviating.

Answer: We agree with the expert reviewer. Dear reviewer, this abbreviation was changed into the exact age group throughout the manuscript text and figures.

Table 2: Place entire table on Page 4 for readability.

Answer: Fixed, as requested.  

Figure 1: Please define the “M” groups for clarity. This defining will probably be more appropriate in the text, as opposed to under the figure description.

Answer: We agree with the expert reviewer. Dear reviewer, this abbreviation was changed into the exact age group throughout the manuscript text and figures. 

Page 4, lines 133 and 134: Be consistent with how interactions are noted. Previously it was without spaces about the “×”; here there are spaces. The latter is encouraged.

Answer: We agree with the expert reviewer. Thank you for the comment. This issue was doubled checked and fixed throughout the manuscript

Table 3: same as Table 2.

Answer: Fixed, as requested.  

Page 5, line 141: Place a space between the Figure 2 description and following text.

Answer: We agree with the expert reviewer and changed as suggested.

Page 5, line 142: Place a space between :100” and “m” for consistency.

Answer: We agree with the expert reviewer and changed as suggested.

Page 6, line 168: Specifically define “running”. Also, please provide references to support the statement.

Answer: We agree with the expert reviewer and we want to define “running” better.

Running is a sport discipline and can be described for example as a triathlon, marathon or a half marathon run in a competition. Therefore, we provided references in our statement.

Page 7, lines 178 – 180: What does the literature say about Type I fibre types; being that they are more aerobic in nature? Also see Page 8, line 223.

Answer: We agree with the expert reviewer and we want to describe the type I fibre types

better. Normal aging is characterized by muscular atrophy due to a decrease in both size and

number of muscle fibers, especially of the type II fibers. In master endurance athletes, type I fiber composition is increased, whereas type II fiber size is reduced. This may have a positive impact on muscle capillarization, oxidative metabolism and therefore endurance performance. With aging sedentary individuals, a common observation of aging endurance athletes is a decreased cross-sectional area of type II fibers while maintaining or increasing the size of their type I fibers as a result of a shift towards a higher expression of myosin heavy chain isoforms of type I fibers. It appears that capillary density and capillary fiber ratio are maintained in athletes who continue endurance training into older age.

Page 7, lines 183 – 184: What is meant by “employed a smaller percentage of their event times”? Please clarify. Same for following statement.

Answer: We agree with the expert reviewer. The study analyzing elite athletes in individual medley has shown that men accomplished a lower race time in breaststroke than did the women and a faster race time in the freestyle in both the 200- and 400-m distances, with the fastest stroke for both sexes being the butterfly.

Page 7, lines 190 – 194: Expound more on the ‘why’ behind this. Yes, it may be considered intuitive, but there are factors (chiefly physiological and psychological factors) that are playing a large role in this observation.

Answer: We agree with the expert reviewer and discussed further this aspect (“This finding might be interpreted considering the different human energy transfer systems engaged in performance as the distance of the race increased [22]. Compared to the shortest distance relied mostlyon anaerobic system, the longer the race, the more the resynthesis of ATP relied on aerobic system; althought the aerobic system allowed swimimming for longer distance, its power was smaller than anaerobic system.”).

Page 7, lines 196 – 200: Again, ‘why’? What are the authors speculations about this observation?

Answer: We agree with the expert reviewer and discussed further this aspect (“An increase of swimming time during the second half of 100 m race indicated a positive pacing. It might be assumed that the accumulated fatigue (increase of blood lactate, metabolic acidosis) caused by high-intensity (predominantly anaerobic) [24] exercise such as the performance in the first half of 100 m race did not allow the maintenance of swimming speed throughout the race leading into an increase of swimming time.”).

Page 7, lines 202 – 208: Same as above.

Answer: We agree with the expert reviewer and discussed further this aspect (“An interpretation of this finding might be that the ability of human body for muscle as a function of time would follow a parabolic (half “U-shape”) rather than a liner relationship. Accordingly, swimming time would increase (i.e., get slower) more in the second lap - compared to the first - than in the last two laps. Considering their physiological relevance, it might be assumed that the second, third and fourth lap (all with high contribution of the aerobic energy transfer system) would have larger affinity among them than with the first lap (relied mostly in the anaerobic system) [26].”).

Page 7, line 210: Replace “further” with “another” for readability.

Answer: We agree with the expert reviewer and changed as suggested.

Page 7, lines 210 – 219: Same as above. What are the authors thoughts on why we are seeing what we are seeing?

Answer: We agree with the expert reviewer and developed further this part (“A low lap-to-lap variability might be considered relevant to a more even pacing, characteristic of high-performance level athletes showing an ability to distribute energy optimally throughout a race. On the other hand, an incorrect distribution of energy might lead athletes of lower performance level either to start fast and show large decrease of speed (positive pacing) or to start slow and increase speed in the last part of a race (final spurt).”). 

Page 8, line 235: Eliminate space and hyphen in “baseline”.

Answer: We agree with the expert reviewer and changed as suggested.

Reviewer 3 Report

IJERPH 799778

In general, the article seems very well written. The Introduction is complete, clear and precise. Appropriate and accurate method of analysis. Objective and precise Discussion, defining the limitations of the study very well. I would just like to suggest a few small improvements in the figures and tables, as I have detailed below.

Results

Line 122 “… M30 and F30 were the fastest age groups for”. - Please put in writing what M30 and F30 mean. The reader should not have to guess abbreviations, which are not really necessary in this case. This also applies to lines 157 and 158 in the Discussion section.

The figures and tables may be slightly improved.

In particular, Figures 1and 2 have the x axis with age, but only the letter "M" appears, which seems to indicate that it is a figure only on the data of men, when in fact it is of both sexes. I suggest two options to make your figure (which is beautiful, congratulations) clearer and more precise: a) On the X axis, remove the letter "M" in front of the number that indicates the age. Another option is to let the X axis of women appear and add the letter F to each number that indicates the age (similar to what you did for men). The first option seems more pleasant, as the figure is "cleaner". It's your choice.

Tables 1 and 2 can be summarized in just one, adding two more columns and separating the speed results from the CV results. It remains the suggestion in favor of using the space of the magazine more economically, without jeopardizing the clarity and precision of the presentation of its results.

As a fine suggestion, figures 3 and 4 may be just one.

Thank you for the opportunity to review this article.

Author Response

Reviewer 3

In general, the article seems very well written. The Introduction is complete, clear and precise. Appropriate and accurate method of analysis. Objective and precise Discussion, defining the limitations of the study very well. I would just like to suggest a few small improvements in the figures and tables, as I have detailed below.

Results

Line 122 “… M30 and F30 were the fastest age groups for”. - Please put in writing what M30 and F30 mean. The reader should not have to guess abbreviations, which are not really necessary in this case. This also applies to lines 157 and 158 in the Discussion section.

Answer: We agree with the expert reviewer. Dear reviewer, this abbreviation was changed into the exact age group throughout the manuscript text and figures.

The figures and tables may be slightly improved.

Answer: Please see response to comments below. 

In particular, Figures 1and 2 have the x axis with age, but only the letter "M" appears, which seems to indicate that it is a figure only on the data of men, when in fact it is of both sexes. I suggest two options to make your figure (which is beautiful, congratulations) clearer and more precise: a) On the X axis, remove the letter "M" in front of the number that indicates the age. Another option is to let the X axis of women appear and add the letter F to each number that indicates the age (similar to what you did for men). The first option seems more pleasant, as the figure is "cleaner". It's your choice.

Answer: We agree that the M might seem confusing and changed the X axis for the range of each age-group. Thank you for the comment.  

Tables 1 and 2 can be summarized in just one, adding two more columns and separating the speed results from the CV results. It remains the suggestion in favor of using the space of the magazine more economically, without jeopardizing the clarity and precision of the presentation of its results.

Answer: We agree with the expert reviewer and merged the tables.

As a fine suggestion, figures 3 and 4 may be just one.

Answer: We merged Fig 3 and 4, as suggested.

Thank you for the opportunity to review this article.